# Metaverse for Exercise Rehabilitation: Possibilities and Limitations

**DOI:** 10.3390/ijerph20085483

**Published:** 2023-04-12

**Authors:** Kyoung-Hwan Cho, Jeong-Beom Park, Austin Kang

**Affiliations:** 1Department of Special Physical Education, Daelim University College, Anyang 13916, Republic of Korea; 2Department of Medicine, Seoul National University, Seoul 08826, Republic of Korea

**Keywords:** metaverse, rehabilitation, Delphi, stroke

## Abstract

Objectives: This study aimed to obtain a consensus agreement from an expert panel on the metaverse for exercise rehabilitation in stroke patients using the Delphi technique. Methods: This study recruited twenty-two experts and conducted three rounds of online surveys between January and February 2023. The Delphi consensus technique was performed online to review and evaluate the framework module. A panel of experts, including scholars, physicians, physical therapists, and physical education specialists in the Republic of Korea, was invited to participate in this study. For each round, the expert consensus was defined as more than 90% of the expert panel agreeing or strongly agreeing with the proposed items. Results: A total of twenty experts completed the three Delphi rounds. First, virtual reality-assisted (VR) treadmill walking could improve cognitive function, concentration, muscular endurance, stroke prevention, proper weight maintenance, and cardiorespiratory function. Second, related technology, safety, price, place, and securing experts would be obstacles or challenges in VR-assisted treadmill walking for stroke patients. Third, the role of exercise instructors in exercise planning, performance, and assessment for VR-assisted treadmill walking is equally important, and reeducation for them is required. Fourth, VR-assisted treadmill walking for stroke patients requires an exercise intensity of at least five times a week, about one hour each time. Conclusions: This study showed that the metaverse for exercise rehabilitation for stroke patients could be successfully developed and would be feasible to be implemented in the future. However, it would have limitations in terms of technology, safety, price, place, and expert factors to be overcome in the future.

## 1. Introduction

A stroke is characterized by pathological, radiological, or clinical evidence of hemorrhage or ischemia affecting a specific cerebral vascular territory [1], which is the leading cause of disability in the elderly and has become one of the major causes of death worldwide [2]. Risk factors include age, obesity, hypertension, diabetes mellitus, elevated blood cholesterol, atrial fibrillation, carotid stenosis, cigarette smoking, poor diet, and physical inactivity [3]. Furthermore, physical inactivity can propagate disability after stroke through physical deconditioning and learned nonuse [4]. It is also reported that stroke can affect the ability to generate force due to a number of mechanisms, including decreased corticospinal drive to the spinal motoneurons, decreased numbers of motor units, transsynaptic degeneration of spinal motor neurons, as well as altered mechanical properties of the muscle that increase joint stiffness [5]. Thus, impairments resulting from a stroke can lead to persistent difficulties with walking and, subsequently, improved walking ability is one of the highest priorities for patients who have experienced a stroke [6]. 

Moreover, the application of exercise to improve motor function can be a major component of stroke rehabilitation. Exercise has been used to increase muscle mass, strength, and physical function in post-stroke patients [7], which is also essential for preventing secondary complications related to cardiovascular-related events, such as a heart attack or a second stroke. As a result, exercise has been extensively used to improve the ability to perform functional motor tasks, and walking is known as a simple means of functional physiotherapy for stroke patients [8]. Although there are various therapeutic methods for helping stroke patients regain their balance and gait, functional electrical stimulation [9], treadmill whole-body vibration training [10], and robot-assisted training [11] have attracted social attention. Moreover, one of the most common methods of improving gait is through overground walking or treadmill walking [12]. Treadmill walking can improve walking speed and distance in ambulatory people after a stroke and is not reported to be inferior to overground walking [13]. 

However, conventional treadmills are neither responsive nor omnidirectional. Therefore, conventional treadmills are used in virtual reality applications, such as for rehabilitation training or gait studies, and VR-assisted rehabilitation therapy is recognized for its potential [14]. As VR is a major component of the metaverse [15], it has become a promising area of health and science research [16]. The term “metaverse” was coined in 1992 by Neal Stephenson in his novel, “Snow Crash” [17], and it refers to a digital universe, which is a perpetual and persistent multiuser environment that combines physical reality and digital virtuality [18]. Technology underpinning the metaverse enables multisensory interactions with virtual environments, digital objects, and humans [19]. Thus, the metaverse is an immersive 3D virtual environment, a true virtual artificial community in which avatars interact with each other as the user’s alter ego [20]. Recently, the metaverse has had its moment in the limelight during the COVID-19 pandemic [21] since people have been physically isolated due to the spread of this infectious disease, which triggered the growth of the metaverse [22]. The metaverse has been considered as a new iteration of the internet that utilizes VR headsets, blockchain technology, and avatars within a new integration of the physical and virtual worlds [23].

Against this background, VR-assisted treadmill walking has been adopted for the rehabilitation of pain and stroke [24]. According to the literature, walking training associated with virtual reality-based training effectively increases walking speed after a stroke [25,26]. However, few studies have analyzed experts’ perceptions of the metaverse for exercise rehabilitation in stroke patients. This study aimed to obtain a consensus agreement from an expert panel on the metaverse for exercise rehabilitation in stroke patients using the Delphi technique, which is a widely used and accepted method for gathering data from respondents or participants within their domain of expertise [27]. In principle, a three-round Delphi method used in this study formed an expert panel, asked questions, synthesized, appraised, communicated feedback, and directed the expert panel to consensus building [28,29,30]. 

## 2. Material and Methods

### 2.1. Study Design

We performed a Delphi study to review and evaluate the metaverse for exercise rehabilitation, with a specific focus on VR-assisted treadmill walking for stroke patients. The Delphi technique is an iterative and anonymous process that can be used to achieve consensus on a subject using multiple rounds of discussion with controlled feedback [31]. Thus, the experts were allowed to change their earlier judgment in each questionnaire iteration. The statistical average was used to represent the experts’ final appraisal. This study was conducted in compliance with the ethical issues presented in the Declaration of Helsinki, according to the guidelines of the Korean government that general surveys are not subject to review by the institutional review board. Figure 1 illustrates the process of the Delphi study [32].

### 2.2. Participants

Since defining an expert is subjective, a panel does not need to be representative of any particular population [33]. Thus, we invited thirty experts to participate in the research via an e-mail with information about the study. The expert panel included scholars, physicians, physical therapists, and physical education specialists in the Republic of Korea. They were selected to provide their opinions and judgments based on their relevant knowledge, expertise, and experiences in exercise rehabilitation for stroke patients. This study selected a panel of participants who had more than five years of experience in their field. As shown in Figure 1, thirty experts were invited, twenty-two participated in the first round, two dropped out due to personal reasons (time constraints), and twenty completed the second and third rounds. Table 1 shows the demographic characteristics of the experts who participated in the Delphi process.

### 2.3. Data Collection

Three Delphi rounds were held approximately between January and February 2023. The first author was assigned as the facilitator who was responsible for identifying and recruiting the expert panel and sending the formal invitation and reminder. Moreover, each Delphi round consisted of data collection, data analysis, and controlled feedback. Experts who understood the purpose of this study and voluntarily agreed to participate in the study conducted an open-ended questionnaire in the first round of the Delphi study. Two weeks before the start of the first round, the moderator contacted and invited expert panelists to participate in this study via e-mail or social media. A brief description of the goals, procedures, and expected timeframe of the overall study was provided via e-mail or social media [32]. An open questionnaire was sent to derive unbiased opinions at the first Delphi round. In the second and third Delphi rounds, the experts were required to rate the importance of all feedback provided by the others to achieve a final consensus on the developed framework module [34]. A summary of feedback from the first Delphi round and a revised framework module were presented to the experts in the second round. For the third Delphi round, a summary of responses from the second Delphi round and a revised framework module were sent to the panelists.

### 2.4. Research Instruments

In the first Delphi round, we collected the panelists’ free opinions on the possibilities and limitations of the metaverse for exercise rehabilitation, focusing on VR-assisted treadmill walking for stroke patients (Figure 2). The concept of VR-assisted treadmill walking was presented to the panelists. In other words, stroke patients could use the safety ring, wear metaverse glasses, and perform walking exercises on a treadmill where conditions suitable for them were automatically set. What differentiates this method from existing treadmills is that it can provide stroke patients with an opportunity to exercise safely with a sense of reality in a virtual space with others. Therefore, it can be expected that stroke patients can have appropriate exercise effects in the virtual world provided beyond the constraints of time and space, but factors such as technical limitations, safety, and price can be presented as limitations.

The second and third Delphi rounds were conducted with a five-point Likert scale (1 = strongly disagree, 2 = disagree, 3 = neutral, 4 = agree, 5 = strongly agree). Table 2 shows the research instruments used in this study. 

In the first round, experts were free to answer the open-ended question, “What would be your opinion on exercise rehabilitation related to the metaverse?” which was reviewed through content analysis. In the second round, experts were asked to present their views on each item using a 5-point Likert scale. Since content validity can be determined by checking the adequacy with which the survey samples, the content, or the objectiveness of the course or area being accessed, we assessed the content validity ratio (CVR), a linear transformation of a proportional level of agreement, which represents the proportion of the participants rating the efficiency of the visual formats according to the explored domains [34]. The panelists were allowed to reconsider their answers to the questionnaires after they heard the answers of the other panelists. In the third round, we re-evaluated the importance and validity of each item by calculating the mean, standard deviation, and CVR using a 5-point Likert scale. As a result, this study could reach a consensus on the main issues.

### 2.5. Data Analysis

All statistical analyses were performed using the IBM Statistical Package for the Social Sciences (SPSS) Version 23.0 statistical software (IBM, Armonk, NY, USA). In this study, consensus could be reached if the coefficient of variation for the panelists was greater than 0 and less than 0.5; however, there was a less-than-satisfactory consensus, and additional investigation could be required if the coefficient was greater than 0.5 and less than 0.8. Consensus was defined as poor and additional investigation would be definitely required if the coefficient was greater than 0.8 [35]. As the coefficients derived for all items were below 0.5 in this study, it could be seen that experts reached a strong degree of consensus. The CVR proposed by Lawshe [36] was determined by using the following formula:$$\mathrm{CVR}=\frac{\mathrm{Ne}-N/2}{N/2}$$

In all three Delphi rounds, the coefficients derived for all items were below 0.5, showing that the panelists reached a strong degree of consensus. The number of panelists determines the final decision to keep the item based on the CVR. Table 3 presents the minimum value of CVR per number of panelists. In this study, the CVR should be higher than 0.42 since the number of panelists was twenty [36], which required a minimum level of agreement among the panelists exceeding 42% [37]. Therefore, items with a CVR ratio of 0.42 or less were excluded, and the contents of each question were ranked by the mean value [34]

## 3. Results

### 3.1. Round 1

In the first Delphi round, we provided basic information for panelists to review, asked them to give their honest feedback, and the facilitator, who was the first author in this study, collected their answers. Table 4 shows open-ended questions presented in the first Delphi round and experts’ response.

The panelists’ key answers in the first Delphi round present that VR treadmill walking could be practical, applicable, feasible, and effective, but there would be barriers or challenges in terms of related technology, safety, price, installation location, and securing experts. In addition, since exercise instructors would play an important role in exercise planning, performance, and evaluation, some panelists suggested that it would be necessary for them to be retrained. However, there were negative opinions about the future feasibility or effectiveness of VR treadmill walking. After the first round of the Delphi survey, the experts were given feedback on the results and asked to adjust or change the rating on each factor [38].

### 3.2. Round 2

In the second Delphi round, we carefully reviewed each answer and found common reoccurrences or themes among all of the answers. Thus, we provided important information from the answers to the panelists, who reviewed the anonymous responses. Panelists were then given the option to readjust their original answers based on what they had just read, which they submitted to the facilitators. Based on the first Delphi round, experts’ opinions were measured on a five-point Likert scale for three domains (eight possibilities, five limitations, and four instructor roles). Table 5 shows the results of the second Delphi round. The mean was the average of the scores given by experts who evaluated each item. Moreover, the standard deviation was the square root of the sum of squared differences from the mean divided by the total number of experts. Meanwhile, CVR was determined in the same way as reviewed previously [36]. In the area of possibility, experts unanimously agreed on the improvement of cognitive function, concentration, muscular endurance, stroke prevention, proper weight maintenance, and cardiorespiratory function. However, experts showed that VR-assisted treadmill walking had no significant effect on strengthening muscle strength (CVR = 0.40) or enhancing agility (CVR = 0.20) of stroke patients. As the CVR was lower than 0.42, they were excluded in the third Delphi round. Moreover, experts unanimously agreed that limitations related to technology, safety, price, place, and securing experts would be obstacles or challenges in VR-assisted treadmill walking (CVR = 1.00). Furthermore, the experts revealed that the exercise instructor’s roles in exercise planning, performance, and assessment were equally important, and all emphasized the need for the retraining of the instructor (CVR = 1.00). 

### 3.3. Round 3

Based on the second Delphi round, experts’ opinions were measured on a five-point Likert scale for four domains (five possibilities, five limitations, four instructor roles, and two exercise intensities). All the experts provided adjusted responses in the third Delphi round. After we reviewed their answers again to find similarities and common themes, we could end this study on the third round because the panelists reached a general group consensus. Table 6 shows the results of the third Delphi round. The mean was the average of the scores given by experts who evaluated each item. In addition, the standard deviation was the square root of the sum of squared differences from the mean divided by the total number of experts. Meanwhile, CVR was determined in the same way as reviewed previously [36]. In the area of possibility, experts reached a unanimous agreement on the improvement of cognitive function, concentration, muscular endurance, stroke prevention, proper weight maintenance, and cardiorespiratory function (CVR = 1.00). As for the limitations, they reached a consensus that related technology, safety, price, place, and securing experts would be obstacles or challenges in VR-assisted treadmill walking for stroke patients (CVR = 1.00). In addition, they revealed that the exercise instructor’s roles in exercise planning, performance, and assessment were equally important, and all emphasized the need for the retraining of the instructor. Regarding the exercise intensity (duration, frequency) for VR-assisted treadmill walking, experts presented a consensus view of at least five times a week, about one hour each time (CVR = 1.00). 

## 4. Discussion

Cirstea [39] reports that nearly 80% of stroke survivors would have initial gait impairment after stroke, of which 15–30% still endure severe gait deficits at six months. Gait deficits after stroke can be caused by several underlying impairments, which include pronounced gait asymmetry, decreased speed, and marked dyscoordination of the paretic limb [40]. It is reported that gait rehabilitation interventions can provide some improvement in the gait pattern [41]. Thus, this study reviewed the metaverse for exercise rehabilitation for stroke patients. In particular, this study analyzed the possibilities and limitations of VR-assisted treadmill walking recognized by experts through the Delphi technique. The Delphi method is the process of collecting opinions on a particular research question, using a series of questionnaires to generate expert opinion in an anonymous fashion, and taking place over a series of rounds [31]. As the Delphi technique has become a popular strategy that straddles both quantitative and qualitative realms, it is also used in health sciences [42]. Therefore, we believed that the Delphi method could be appropriate for analyzing experts’ opinions about the possibilities and limitations of metaverse rehabilitation exercises that have not yet been realized in our society.

In the Delphi method, the experts’ responses can be all anonymous [43]. After each round, facilitators need to review and sort through all the answers. In addition, facilitators are often required to locate answers with common themes and ideas, sharing these with the other experts. Once the experts know the other panelists’ answers, they can adjust their answers according to the group’s responses [44]. Since the answers are all anonymous, panelists who participate in the Delph rounds can feel more comfortable providing their honest answers without receiving judgments. The Delphi method is considered as an efficient procedure for generating knowledge [45].

Considering this Delphi method, we conducted this study as follows. In the first round, we provided basic information for panelists to review on the metaverse rehabilitation exercise, asked them to give their honest feedback, and the facilitator collected their answers. In the second Delphi round, we carefully reviewed each answer and found common reoccurrences or themes among all of the answers. Thus, we provided important information from the answers to the panelists, who reviewed the anonymous responses. Panelists were then given the option to readjust their original answers based on what they had just read, which they submitted to the facilitators. In the third Delphi round, all the experts provided adjusted responses. After we reviewed their answers again to find similarities and common themes, we could end this study because the panelists reached a general group consensus.

Thus, the following results could be derived through three Delphi rounds. First, VR-assisted treadmill walking could improve cognitive function, concentration, muscular endurance, stroke prevention, proper weight maintenance, and cardiorespiratory function. Second, related technology, safety, price, place, and securing experts would be obstacles or challenges in VR-assisted treadmill walking for stroke patients. Third, the role of exercise instructors in exercise planning, performance, and assessment for VR-assisted treadmill walking is equally important, and reeducation for them is required. Fourth, VR-assisted treadmill walking for stroke patients requires an exercise intensity of at least five times a week, about one hour each time. Zhang et al. [46] report that VR interventions effectively improve upper- and lower-limb motor function, balance, gait, and daily function of stroke patients, but have no benefits on cognition, which partially supports the findings of this study. However, we did not directly interview or analyze stroke patients. In addition, the Delphi study can prioritize expert opinions, but experts’ opinions may not necessarily be the best choice because excessive criticism and optimism may be shown, and data manipulation may occur. Since the credibility of the resulting recommendations can depend on the rigorous use of the Delphi technique, there is a need for consistency and quality in conducting and reporting studies, such as guidance on Conducting and REporting of DElphi Studies (CREDES) [47]. However, this study has a limitation in that the Delphi technique according to CREDES was not implemented. Therefore, follow-up research needs to conduct the Delphi technique considering CREDES targeting various participants.

## 5. Conclusions

This study conducted three Delphi rounds to obtain a consensus agreement from an expert panel on the metaverse for exercise rehabilitation in stroke patients. A total of twenty experts who participated in this study reached the following consensus. There would be the possibility that metaverse for rehabilitation exercise could help stroke patients improve cognitive function, concentration, muscle endurance, and cardiorespiratory function, as well as maintain proper weight, while it could prevent the recurrence of stroke. Moreover, the metaverse rehabilitation exercise for stroke patients would have limitations regarding technology, safety, price, place, and expert factors. As a result, the metaverse for exercise rehabilitation for stroke patients could be successfully developed and would be feasible to be implemented. In particular, it was confirmed that experts recognized that VR-assisted treadmill walking had various exercise effects on stroke patients. They revealed that VR-assisted treadmill walking would be a helpful exercise for stroke patients if technical development could be achieved and appropriate price and safety issues could be solved. Exercise instructors play an important role in exercise planning, performance, and evaluation using VR-assisted treadmill walking, so retraining is required. However, this study has a limitation in that no empirical analysis was conducted on patients who performed VR-assisted treadmill walking, but it has significance in that it reviewed experts’ perceptions of rehabilitation treatment using the metaverse.

## Figures and Tables

**Figure 1 ijerph-20-05483-f001:**
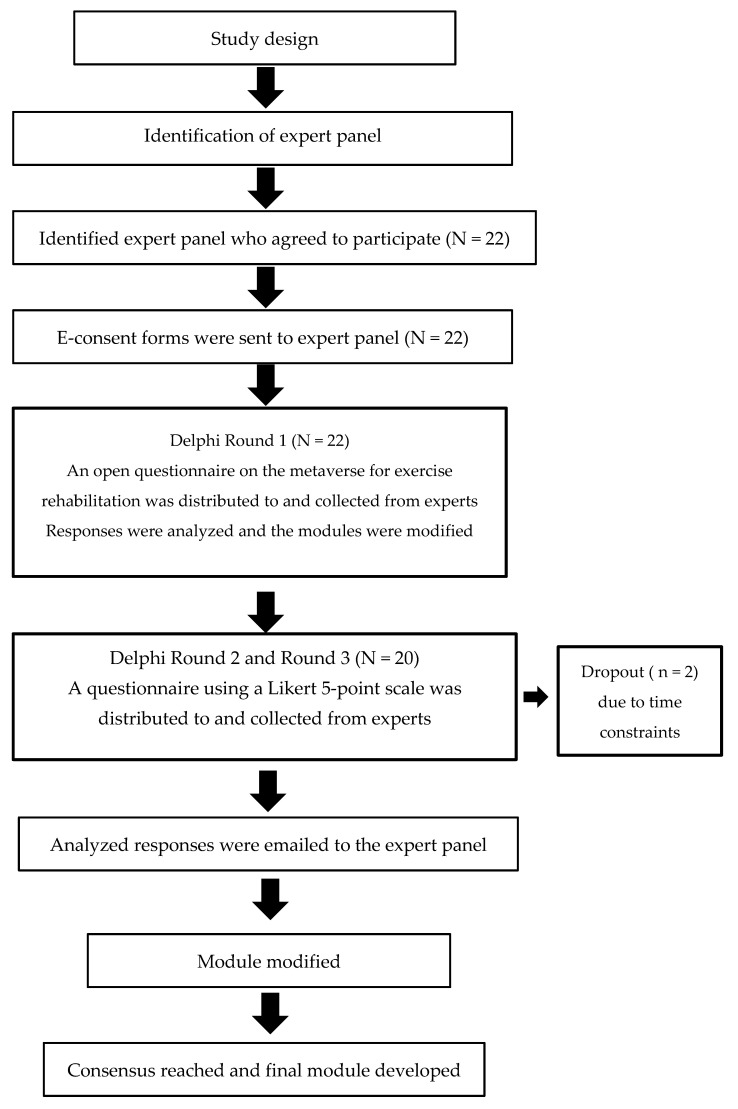
The process of the Delphi study [32].

**Figure 2 ijerph-20-05483-f002:**
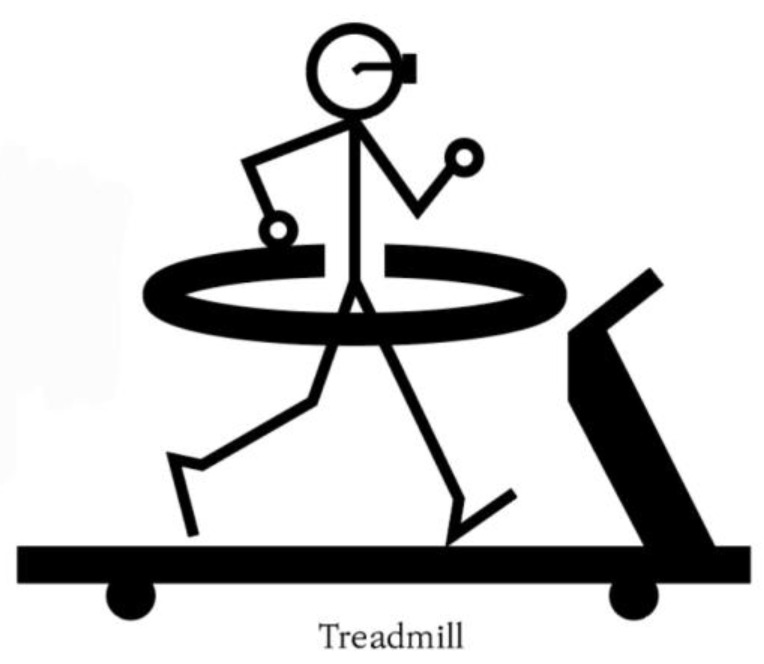
A conceptual diagram of a metaverse-assisted treadmill with a safety ring.

**Table 1 ijerph-20-05483-t001:** The demographic characteristics of the expert panel.

No.	Age (Years)	Gender	Career (Years)	Occupation	Note
1	51	M	18	Scholar	Physical education
2	49	M	7	Scholar	Physical education
3	42	M	8	Scholar	Physical education
4	46	M	10	Physician	Rehabilitation medicine
5	38	F	8	Physician	Rehabilitation medicine
6	48	M	13	Physician	Rehabilitation medicine
7	43	M	10	Physical therapist	
8	39	F	8	Physical therapist	
9	50	M	15	Physical therapist	
10	45	M	11	Physical therapist	
11	44	F	10	Physical therapist	Dropped out
12	38	M	9	Physical education specialist	
13	36	F	6	Physical education specialist	
14	38	M	9	Physical education specialist	
15	40	M	10	Physical education specialist	
16	41	M	11	Physical education specialist	
17	42	M	11	Physical education specialist	
18	46	M	16	Physical education specialist	
19	39	M	10	Physical education specialist	
20	36	F	8	Physical education specialist	
21	41	M	11	Physical education specialist	Dropped out
22	44	M	15	Physical education specialist	

**Table 2 ijerph-20-05483-t002:** Research instruments.

Round	Instruments	Content	Analysis Tool
Round 1	Open-ended questions	Initial views on the metaverse for exercise rehabilitation	Content analysis
Round 2	5-point Likert scale	Content analysis used to determine the importance of specific items in each area.	Mean and quadrant range content feasibility, and CVR *
Round 3	5-point Likert scale	Reassessment of importance and feasibility for each item	Mean and quadrant range content feasibility, and CVR *

* CVR = Content validity ratio.

**Table 3 ijerph-20-05483-t003:** Minimum value of CVR (content validity ratio).

Category	Value for 10 or Less						Value for 30 or Less		
Number of panelists	5	6	7	8	9	10	15	20	30
Minimum value	0.99	0.99	0.99	0.75	0.68	0.62	0.49	0.42	0.33

Source: Lawshe [36].

**Table 4 ijerph-20-05483-t004:** Results of Delphi Round 1.

Questions	Component	Key Answers
What does the metaverse for exercise rehabilitation include? (possibilities)	Practicality	VR-assisted treadmill walking could be practical.
	Applicability	VR-assisted treadmill walking could be applicable.
	Feasibility	VR-assisted treadmill walking could be feasible.
	Effectiveness	VR-assisted treadmill walking could be effective.
What does the metaverse for exercise rehabilitation include? (limitations)	Challenges/Barriers	Limitations include the technology, safety, price, place, and securing of experts for VR-assisted treadmill walking.
What is the exercise instructor’s role in the metaverse for exercise rehabilitation?	Exercise instructor’s role	When stroke patients perform VR-assisted treadmill walking, instructors play an important role in exercise planning, guidance, and evaluation for them. Therefore, reeducation and training of instructors are also required.

**Table 5 ijerph-20-05483-t005:** Results of Delphi Round 2.

Domains	Content	Mean	Standard Deviation	Content Validity Ratio
Possibilities	Cognitive function	5.00	0.00	1.00
	Concentration	4.95	0.04	1.00
	Muscular endurance	4.9	0.04	1.00
	Prevention of stroke recurrence	4.90	0.07	1.00
	Maintaining an appropriate weight	4.90	0.07	1.00
	Cardiopulmonary function	4.85	0.11	1.00
	Strengthening of muscles	3.95	0.74	0.40
	Improvement of agility	3.60	0.42	0.20
Limitations	Technology	5.00	0.00	1.00
	Safety	4.95	0.04	1.00
	Price	4.95	0.04	1.00
	Place	4.70	0.21	1.00
	Experts	4.30	0.21	1.00
Exercise instructors	Planning	5.00	0.00	1.00
	Implementation	5.00	0.00	1.00
	Assessment	5.00	0.00	1.00
	Retraining	4.90	0.07	1.00

**Table 6 ijerph-20-05483-t006:** Results of Delphi Round 3.

Domains	Content	Mean	Standard Deviation	Content Validity Ratio
Possibilities	Cognitive function	5.00	0.00	1.00
	Concentration	5.00	0.00	1.00
	Muscular endurance	5.00	0.00	1.00
	Prevention of stroke recurrence	5.00	0.00	1.00
	Maintaining an appropriate body weight	5.00	0.00	1.00
	Cardiopulmonary function	5.00	0.00	1.00
Limitations	Technology	5.00	0.00	1.00
	Safety	5.00	0.00	1.00
	Price	5.00	0.00	1.00
	Place	5.00	0.00	1.00
	Experts	4.90	0.07	1.00
Exercise instructors	Planning	5.00	0.00	1.00
	Implementation	5.00	0.00	1.00
	Assessment	5.00	0.00	1.00
	Retraining	4.90	0.07	1.00
Exercise intensity	1 h or less	5.00	0.00	1.00
	5 or more times a week	4.70	0.21	1.00

## Data Availability

Not applicable.

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
