# Peer review of "Metaverse for Exercise Rehabilitation: Possibilities and Limitations"

_ijerph, 2023, doi:10.3390/ijerph20085483_

Round 1

Reviewer 1 Report

This study aimed to obtain a consensus agreement from an expert panel on the metaverse for exercise rehabilitation in stroke patients using the Delphi technique. Through three rounds of Delphi to reach a consensus among experts, this study showed that the metaverse for exercise rehabilitation for stroke patients could be successfully developed and feasible to be implemented using the Delphi technique. In conclusion, I think it needs minor revision to enhance the scientific level. More detailed comments or suggestions are listed below.

1.      There are some minor grammatical errors and typos. The paper needs to be revised in terms of language.

2.      The paper mentions that the user's intention is solved according to the relevant algorithm. In the revision, the author should make some methodological elaboration and explanation for the part of the original technology cited by others, and then apply the technology to your own problems.

3.      The article lacks a detailed description of Delphi study.

4.      Figure 2 needs to be introduced more so that readers can understand its system configuration.

5.      More details need to be provided in the data analysis section of Table 4 and Table 5. For example, the calculation process of Standard De-viation, mean.

Author Response

Dear Reviewer
We revised our manuscript as follows;

  1. There are some minor grammatical errors and typos. The paper needs to be revised in terms of language.
    -->We revised our paper in terms of language.
  2. The paper mentions that the user's intention is solved according to the relevant algorithm. In the revision, the author should make some methodological elaboration and explanation for the part of the original technology cited by others, and then apply the technology to your own problems.
    -->In the revision, we derived our solution through three Delphi rounds focusing on the possibilities and limitations of metaverse rehabilitation exercise.
  3. The article lacks a detailed description of Delphi study.
    -->We described detailed procedures for each round by adding them to the methodology.
  4. Figure 2 needs to be introduced more so that readers can understand its system configuration.
    -->We added some more explanation to the concept of metaverse exercise rehabilitation presented in Figure 2.
  5. More details need to be provided in the data analysis section of Table 4 and Table 5. For example, the calculation process of Standard De-viation, mean
    -->We described the mean, SD, and CVR calculation methods presented in Tables 4 and 5 in more detail.

Reviewer 2 Report

It is an interesting article that talks about the metaverse as a possibility in physical recovery after a heart attack. The subject is new and opens the possibility of new virtual therapies.

However, although the methodology is well written and the results show what the proposed three rounds of exercises reflect, the discussion is very weak. It only has 4 references and presents results again. I suggest redoing this part and focusing on whether the opinion of the experts is valid or not to determine if the metaverse is a good therapeutic alternative for heart attack patients.

Author Response

However, although the methodology is well written and the results show what the proposed three rounds of exercises reflect, the discussion is very weak. It only has 4 references and presents results again. I suggest redoing this part and focusing on whether the opinion of the experts is valid or not to determine if the metaverse is a good therapeutic alternative for heart attack patients.

à We respect the reviewer's views, adding 10 references in the discussion of the results and increasing the volume as follows.

Cirstea [39] reports that nearly 80% of stroke survivors would have initial gait impairment after stroke, of which 15–30% still endure severe gait deficits at six months. Gait deficits after stroke can be caused by several underlying impairments, which include pronounced gait asymmetry, decreased speed, and marked dyscoordination of the paretic limb [40]. It is reported that gait rehabilitation interventions can provide some improvement in the gait pattern [41]. Thus, this study reviewed the metaverse for exercise rehabilitation for stroke patients. In particular, this study analyzed the possibilities and limitations of VR-assisted treadmill walking recognized by experts through the Delphi technique. The Delphi method is the process of collecting opinions on a particular research question, using a series of questionnaires to generate expert opinion in an anonymous fashion, and taking place over a series of rounds [42]. As the Delphi technique has become a popular strategy that straddles both quantitative and qualitative realms, it is also used in health sciences [43]. Therefore, we believed that the Delphi method could be appropriate for analyzing experts' opinions about the possibilities and limitations of metaverse rehabilitation exercises that have not yet been realized in our society.

In the Delphi method, the experts' responses can be all anonymous[44]. After each round, facilitators need to review and sort through all the answers. In addition, facilitators are often required to locate answers with common themes and ideas, sharing these with the other experts. Once the experts know the other panelists' answers, they can adjust their answers according to the group's responses [45]. Since the answers are all anonymous, panelists who participate in the Delph rounds can feel more comfortable providing their honest answers without receiving judgments. The Delphi method is considered as an efficient procedure for generating knowledge [46]

Considering this Delphi method, we conducted this study as follows. In the first round, we provided basic information for panelists to review on the metaverse rehabilitation exercise, asked them to give their honest feedback, and the facilitator collected their answers. In the second Delphi round, we carefully reviewed each answer and found common reoccurrences or themes among all of the answers. Thus, we provided important information from the answers to the panelists, who reviewed the anonymous responses. Panelists were then given the option to readjust their original answers based on what they had just read, which they submitted to the facilitators. In the third Delphi round, all the experts provided adjusted responses. After we reviewed their answers again to find similarities and common themes, we could end this study because the panelists reached a general group consensus.

Thus, the following results could be derived through three Delphi rounds. First, VR-assisted treadmill walking could improve cognitive function, concentration, muscular endurance, stroke prevention, proper weight maintenance, and cardiorespiratory function. Second, related technology, safety, price, place, and securing experts would be obstacles or challenges in VR-assisted treadmill walking for stroke patients. Third, the role of exercise instructors in exercise planning, performance, and assessment for VR-assisted treadmill walking is equally important, and reeducation for them is required. Fourth, VR-assisted treadmill walking for stroke patients requires an exercise intensity of at least five times a week, about one hour each time. Zhang et al. [7] report that VR interventions effectively improve upper- and lower-limb motor function, balance, gait, and daily function of stroke patients, but have no benefits on cognition, which partially supports the results of this study. However, we did not directly interview or analyze stroke patients. In addition, the Delphi study can prioritize expert opinions, but experts' opinions may not necessarily be the best choice because excessive criticism and optimism may be shown, and data manipulation may occur. Since the credibility of the resulting recommendations can depend on the rigorous use of the Delphi technique, there is a need for consistency and quality in conducting and reporting studies, such as guidance on Conducting and REporting of DElphi Studies (CREDES) [48]. However, this study has a limitation in that the Delphi technique according to CREDES was not implemented. Therefore, follow-up research needs to conduct the Delphi technique considering CREDES targeting various participants.

Round 2

Reviewer 2 Report

Well done!!!